# Conversational grounding in emergent communication – data and divergence

**Oliver Lemon** [*]
Interaction Lab and Alana AI
Heriot-Watt University
Edinburgh, Scotland
{o.lemon}@hw.ac.uk

## Abstract

We argue for a new research direction in emergent communication, combining work on 'Conversational Grounding' with Symbol Grounding (SG+CG). We first present the fine-grained and *targeted* feedback signals provided by Conversational Grounding messages and discuss the potential advantages of such a combination. We argue that a key factor holding back research in this area is lack of appropriate data, where *divergent* agents can *resolve disagreements and errors*, and we propose requirements and methods for new data collections enabling such work.

## 1 Introduction

This paper argues for a closer collaboration between the fields of emergent communication and conversational AI research. We discuss the potential advantages of this unified view, and propose new data collections that we will need to enable it, focussed on agent *divergence*.

In human communication, conversational 'grounding' dialogue acts, like repair ('Not that bowl, the white one') and clarification ('Is it white or brown?') are used to establish mutually agreed common ground when engaged in a collaborative task (Brennan & Clark, 1996). Recently, a number of position and survey papers (Schlangen, 2019; Chandu et al., 2021; Benotti & Blackburn, 2021) highlight the importance of these types of *targeted* communicative feedback signals in human task coordination and language learning. Such signals are also likely to be useful for models of emergent communication – especially if we ultimately want communicating agents to be able to coordinate with humans. Table 1 gives brief examples.

Table 1: Examples of Conversational Grounding messages

| | |
|---|---|
| 1) confirmation ("is it X?"); | 2) clarification ("is it X or Y?") |
| 3) other-repair ("no not X but Y") | 4) self-repair ("I mean X not Y") |
| 5) underspecified repair ("no that is not X") | 6) implicit repair "the *bowl* is X." |

As well as being more human-understandable, conversational grounding phenomena could also lead to better task completion, better learned representations (for example in terms of compositionality (Suglia et al., 2020)), and might also help to counter language drift in agent communities. We will discuss such potential advantages, as well as requirements on new data collections needed to train such systems. The paper closes with some future research directions.

## 2 Grounding in conversation versus symbol grounding

Current methods for developing collaborative and communicating AI agents which are situated in the real world focus on *symbol grounding* – see the image in figure 1. Here, emergent communication is often modelled as agents learning how to agree on targets in a visual scene (for example the brown

---

[*]http://sites.google.com/site/olemon/

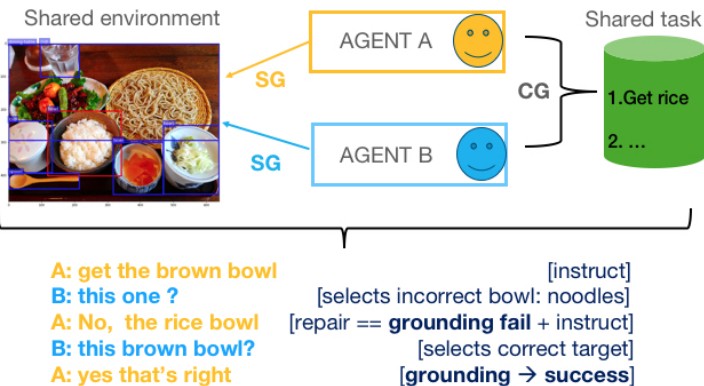

Figure 1: Collaboration = Symbol Grounding (SG) + Conversational Grounding (CG).

bowl of rice), in tasks such as GuessWhat?! (De Vries et al., 2017). To achieve this, an omniscient Teacher agent instructs a Learner about the one true state of the scene and how to properly describe it in a formal or Natural Language (a form of Supervised Learning). But as well as being slow and requiring large amounts of data, this supervised approach cannot account for the ways in which *collaborative meaning is contextual and dynamic, local to a task, co-constructed, and negotiated.*

Consider for example the dialogue in figure 1. As this example illustrates, humans are able to learn and adapt meanings for specific tasks, so what makes communication vital for collaboration is that it allows *divergent* agents to be flexible and adaptive enough to *rapidly reach agreement that is fit for current purposes*, to *adapt* their language when confronted with other agents and other tasks, and to *learn* new concepts. We call these processes, studied in Cognitive Science and models of conversation, *conversational grounding* (CG), rather than *symbol grounding* (SG). They have been argued to be universal in human languages and a foundation of cooperative communication (Brennan & Clark, 1996; Dingemanse et al., 2015; Healey et al., 2018; Benotti & Blackburn, 2021).

In contrast, 'emergent language' agents develop their own communication protocols which are generally not human-understandable (Kottur et al., 2017; Lazaridou & Baroni, 2020). Moreover, emergent communication work does not deal with fine-grained targeted messages such as clarification and repair which are essential to human collaboration, and which might also be beneficial for computational models of emergent communication (see below). This issue is partly related to the restricted nature of the data and tasks used in emergent communication, which are generally simple reference games such as GuessWhat?! using shared images (De Vries et al., 2017; Suglia et al., 2021), where *agents are not divergent* and phenomena such as repair do not occur (as we discuss in section 4), meaning that models for CG tasks cannot yet be trained.

## 2.1 CONVERSATION, COLLABORATION, AND DIVERGENCE

Current work in collaborative AI (Dafoe et al., 2021) uses a variety of *pre-trained Deep Learning models* with general vision and language capabilities, which are then fine-tuned for specific tasks. But such agents will not agree on everything, and currently they cannot detect or recover from disagreement or errors. This is a critical limitation, which prevents the latest neural models from being useful for developing collaborative agents. Sources of potential *divergence* are many – agents' perceptions and language may have been fine-tuned differently, or may have diverged due to continual learning, or they may have different visual perspectives on the current task, simply due to different physical locations, and they may have different plans. If one of the collaborating agents is human, divergences might also be individual (e.g. due to disability), idiosyncratic, or cultural in nature.

We argue that what is needed are new mechanisms for divergent agents to rapidly reach agreement on shared tasks – with each other and with humans – when they may have different perspectives, perceptions, language, and plans. But we already have such mechanisms in everyday global use: conversational grounding in Natural Language. Therefore, we propose a new focus on divergent agent tasks, which combine symbol grounding and conversational grounding: SG+CG

## 3   POTENTIAL ADVANTAGES

Firstly, divergent agents which have no way to reach agreement are unable to collaborate – thus the fundamental potential advantage of SG+CG is that divergent agents will be able to complete a much wider range of shared tasks than current state-of-the-art agents – which are able to learn visually grounded language but cannot perform conversational collaborative grounding.

Secondly, conversational grounding might also be also more computationally effective than symbol grounding alone, as it provides new fine-grained feedback signals for machine learning, rapid adaptation, and optimisation.

Thirdly, the more detailed feedback signals provided by CG might lead to better quality of learned representations, for example perhaps enhancing compositionality. This is because, for example, specific properties of objects can be clarified and repaired, rather than whole class labels, leading to informative model updates. Finally, other issues in emergent communication, such as language drift in communities of agents, might be impacted by CG abilities, as they provide agents with more targeted abilities to correct language use, concepts, and plans of others.

In summary, we could expect quantitative performance gains in terms of 1) ability to coordinate successfully (i.e. task success), 2) speed of learning and adaptation to new tasks, 3) the quality of learned neural representations, for example as determined using the CompGuessWhat!? evaluation framework (Suglia et al., 2020), and 4) properties of emergent languages in communities such as reduced language drift.

## 4   DATA COLLECTION REQUIREMENTS FOR SG+CG

Training data for conversational grounding phenomena such as clarification and repair is missing from all the large-scale vision-and-language learning datasets (Schlangen, 2019; Benotti & Blackburn, 2021; Chandu et al., 2021). Current setups either 1) do not collect *any data at all* on collaborative *grounding phenomena* such as clarification and repair for *divergent agents*, or else 2) do not collect sufficient *volume* of such data; or 3) do not collect *'ecologically-valid'* data in scenarios which are close to real-world tasks. New datasets are needed.

Recent years have seen many different shared tasks and associated datasets for multimodal language learning (to name a few: GuessWhat?! (De Vries et al., 2017), BURCHAK (Yu et al., 2017b), Minecraft (Narayan-Chen et al., 2019), CUPS (Loáiciga et al., 2021), CerealBar (Suhr et al., 2019), IGLU (Kiseleva et al., 2021), TEACh (Padmakumar et al., 2021)). However, most such tasks fail to meet the requirements of SG+CG since they do not focus on collaborative grounding for divergent agents, and/or use only abstract shapes and images (Zarrieß et al., 2016; Yu et al., 2017b; Narayan-Chen et al., 2019; Suhr et al., 2019; Kiseleva et al., 2021) rather than real images or 3D scenes.

Very often the data collection environments do not contain different agent perspectives or perceptions, nor multiple objects of the same type (so no clarification is needed for task success), and/or the tasks do not allow agents to express repairs or clarification (see GuessWhat?!). The few datasets which do meet most of the requirements for SG+CG (e.g. CUPS, TEACh) fail to contain sufficient examples of miscommunication, repairs, semantic coordination etc required for model training.

What is needed is a new focus on CG for divergent agents in ecologically-valid environments such as AI2-THOR and VirtualHome (Kolve et al., 2019; Huang et al., 2022) – where we can create new large-scale collections of conversational grounding phenomena. New online data-collection tools such as SLURK (Götze et al., 2022), which allows multiple humans to communicate about controlled task environments, will also be useful here. Most recently, the TEACh (Padmakumar et al., 2021) dataset used in the 2022 Alexa SimBot challenge provides a set of 3000 human-human dialogues about shared tasks in a simulated 3D home (using the AI2-THOR simulator Kolve et al. (2019)), where a Commander interacts with a Follower. The Commander has egocentric 3D views from both agents, and oracle access to a map and task details. Crucially, the Follower can make mistakes which need correction, and the agents have different perspectives, leading to a limited form of divergence. This setup and environment is closer to what is needed for work on SG+CG

### 4.1 METHODS FOR CREATING DIVERGENCE, DISAGREEMENT, AND RESOLUTION

We will need to use data-collection techniques for divergent agents which can *resolve disagreements* – i.e. which can coordinate different perceptions, language, and knowledge, as was done in the seminal work on Maptask (Anderson et al., 1991). We can use similar ideas to create datasets where agents need to detect and correct ambiguity and disagreements via interaction. As Schlangen (2019) notes, current data sets such as GuessWhat?! are not useful here because for the human crowdworkers looking at the images *" ... the perceptual task being so easy for them, a need for dealing with miscommunication never arose ... and hence no such strategies can be learned from that data."* He proposes the MeetUp game (Ilinykh et al., 2019), where 2 players have to coordinate to meet in a particular place which they must agree on as being the target, and a variant MatchIt (without navigation) where the aim is for 2 agents to decide if they are looking at the same image. Shekhar et al. (2017) create a version of an image captioning dataset (FOIL-COCO) where mistakes (perceptual divergence) are introduced into the original caption (and they showed that state-of-the-art vision and language models could not detect and correct such mistakes). The BURCHAK dataset (Yu et al., 2017b) collected repairs and corrections in an abstract teaching game, where Reinforcement Learning was used to train a grounded language learner for colours and shapes (Yu et al., 2017a). Healey et al. (2018) use a chat tool (DiET) which allows edits and manipulations of dialogue turns, to collect data on handling of disagreements. Each of these efforts collects important collaborative phenomena around agent divergence, but did not collect sufficient volume of data for model training.

Several other data collections such as REX and PentoRef (Tokunaga et al., 2012; Zarrieß et al., 2016) also focus on controlled settings where images are of abstract shapes rather than real-world, but nevertheless collected some data on phenomena such as repair in a joint task setting. The recent IGLU task (Kiseleva et al., 2021) also collects some clarification data, but also focusses on collaborative building of minecraft-style block structures, rather than realistic tasks. In terms of more realistic environments, in the CUPS corpus (Loáiciga et al., 2021) two agents have different views on a simulated tabletop scene, but where some different cups have been removed from each participant's view. Again, this setup elicits some repair and clarification phenomena of the type we target. All of these designs and experiences are useful for collecting SG+CG data, as they exemplify methods with which the data of interest can be collected.

### 4.2 PROPOSAL: DIVERGENCE IN FUTURE SG+CG DATA COLLECTIONS

We propose to create new tasks similar to TEACh (using the open source AI2-THOR) but where we carefully control each agent's/human's *divergent perspectives, perceptions, language, and plans* relevant to a shared task. The *perspectives* will mean that different objects are in view for each agent, divergent *perceptions and language* will involve different class and property labels, and different *plans* will require negotiation. Finally some scenes must also contain distractors (e.g. several different bowls, cups etc) - all of which will lead to a greater volume of repairs, clarification, and negotiation. We can then create specific data-collections which focus on particular SG+CG tasks, for instance clarification and repair of object class/properties, repair of reference, plan repair. Given sufficient data collected in this way, we can then train models for SG+CG in a multi-task fashion.

In summary, learning from previous work such as MapTask, Cups, FOIL, MeetUp, and TEACh, we propose to create new *divergent* Human-Human and Human-Agent data collections with greatly increased environmental pressure (Choi et al., 2018) to perform conversational grounding, for example involving distractors, ambiguity, vagueness, and different agent perspectives leading to disagreement and resolution (Schlangen, 2019; Benotti & Blackburn, 2021; Anderson et al., 1991). Finally, the simulated tasks should ideally have physical counterparts, so that communication trained from data collected in simulation can ultimately be tested in real-world scenarios. Approaches such as RoboTHOR (Deitke et al., 2020), built on AI2-THOR, are well-suited to this requirement.

## 5 RESEARCH DIRECTIONS

The main problem we have discussed in this paper is that current vision-and-language data sets do not allow us to learn conversational grounding for divergent agents. The new data collections outlined above will enable several other important research directions:

- Developing and training new multi-task models of emergent communication which are capable of understanding and generating conversational grounding inputs and outputs.

- Developing tools for the analysis of conversational grounding behaviours and policies (i.e. in what circumstances does grounding occur? What sequences of actions are effective?)

- Evaluating the benefits of SG+CG in real-world tasks where teams of agents (sometimes including humans) need to coordinate on shared tasks, for instance via RoboTHOR.

ACKNOWLEDGEMENTS

Thanks to Alessandro Suglia and Ioannis Konstas for helpful discussions. This work is partially supported by the European Commission under the Horizon 2020 framework programme for Research and Innovation (H2020-ICT-2019-2, GA no. 871245), SPRING project, http://spring-h2020.eu

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
