# OpenReview forum: "Conversational grounding in emergent communication --  data and divergence"
_ICLR.cc/2022/Workshop/EmeCom — EmeCom Workshop at ICLR 2022_

### Official Review · Reviewer_Cm2q · 2022-03-18
**Interesting proposal to focus on new data creation processes to handle disagreement in emecomm research**

**Rating:** Accept
**Confidence:** 4

**Review:**

## SUMMARY

In this paper, the authors argue that current research on emergent communication lacks a fundamental features that is instead ubiquitous in human communication, namely the ability to handle and overcome disagreement. After a thorough review of recent work on environments and datasets for conversational grounding, the authors propose to create a new interactive and grounded datasets through the open source AI2-THOR and similar to TEACh, with the additional feature of having ambiguity and divergence of perspectives, leading to the task of handling, and potentially overcoming, disagreement between learning agents.

## STRENGHTS
One of the key features of human language is that it allows coordination with an unlimited number of speakers to carry out an unbounded set of tasks. This is achieved, amongst others, through the ability to compose known meanings (compositionality), productivity (the ability to generate new linguistic units) but also with the ability to handle disagreement and contextual interactions. Here an important role is played by pragmatics and by speaker/hearer's intentions that are be dynamic and variable in nature. I believe that a call for action to focus on this aspects in emecom research, as argued in the paper, is of great importance for making progress in the field.

## WEAKNESSES
I do not find major weakness in this work. The problem presented is clear, the literature review is exhaustive in breadth but not so much in depth. I would have preferred a bit less coverage in favor of a deeper explanation of some data/methods reported. This is especially true for  TEACh since it is very similar to what the authors proposed that it should be done to address the problem of divergence


## CONCLUSION
In general, I find this paper to be a great fit for the workshop. I believe it can sparkle interesting discussions on the role that current datasets and environments play in emergent communication research and could aid highlighting future directions.

---

### Official Review · Reviewer_M2Q8 · 2022-03-22
**Review of "Conversational grounding in emergent communication -- data and divergence"**

**Rating:** Accept
**Confidence:** 4

**Review:**

## Summary
This paper presents a compelling research direction that I believe will interest the participants at the Emergent Communication workshop. It highlights the difference between "symbol grounding" and "conversational grounding", and persuasively argues for the importance of this distinction in collaborative communicative settings. The authors then point towards collecting data specifically targeted at measuring aspects of conversational grounding as the key first step in making progress towards training AI systems with these capabilities.

Before introducing their data collection proposal, the paper gives a comprehensive, yet efficient, overview of other datasets and clearly discusses their limitations for the problem at hand. The authors then outline their proposed data set, which should be generated to explicitly induce conversational grounding, e.g. through putting humans in situations where communication is hard, and thus fails and needs repair.

## Criticisms
- The authors write: *"We argue that what is needed are new mechanisms for divergent agents to rapidly reach agreement on shared tasks – with each other and with humans – when they may have different perspectives, perceptions, language, and plans. But we already have such mechanisms in everyday global use: conversational grounding in Natural Language."*
	- The part here about agents with diverging languages being able to resolve their differences with natural language is confusing and seems untrue. This claim would need justification.

- The claims that CG might lead to more efficient learning and higher quality representations could use additional justification. In Figure 1, we see that B's confusion resulted in A revealing more information about the grounding of "the brown bowl" in the first sentence. It would be useful if the authors could illustrate their hypotheses in terms of this example to make their cases more compelling.

- To a new reader on "conversational grounding" it is a bit unclear if (or to what extent) the concept is unique to this paper. The authors write in Section 2, paragraph 2, *"We call these processes, studied in Cognitive Science and models of conversation, conversational grounding (CG)"*. This confusion could potentially be alleviated by instead writing, *"These processes are called conversational grounding (CG), and are studied in Cognitive Science and models of conversation"*.

- In the first two sections, the authors make repeated reference to "agent divergence", but the idea is not given any preliminary definition to aid readers until the full definition.

- In the bibliography, the URL for the paper "Running Repairs..." by Healy et al. is broken.

---

### Decision · Program_Chairs · 2022-03-25

**Decision:**

Accept

**Comment:**

Both reviewers agree this is a strong submission and gives a novel perspective on the issue of conversational grounding and disagreement. We look forward to discussions!